# Structure and transport mechanism of P5B-ATPases

Ping Li[1,3], Kaituo Wang [2,3], Nina Salustros[2], Christina Grønberg[2] & Pontus Gourdon [1,2✉]

In human cells, P5B-ATPases execute the active export of physiologically important polyamines such as spermine from lysosomes to the cytosol, a function linked to a palette of disorders. Yet, the overall shape of P5B-ATPases and the mechanisms of polyamine recognition, uptake and transport remain elusive. Here we describe a series of cryo-electron microscopy structures of a yeast homolog of human ATP13A2-5, Ypk9, determined at resolutions reaching 3.4 Å, and depicting three separate transport cycle intermediates, including spermine-bound conformations. Surprisingly, in the absence of cargo, Ypk9 rests in a phosphorylated conformation auto-inhibited by the N-terminus. Spermine uptake is accomplished through an electronegative cleft lined by transmembrane segments 2, 4 and 6. Despite the dramatically different nature of the transported cargo, these findings pinpoint shared principles of transport and regulation among the evolutionary related P4-, P5A- and P5B-ATPases. The data also provide a framework for analysis of associated maladies, such as Parkinson's disease.

[1] Department of Experimental Medical Science, Lund University, Lund, Sweden. [2] Department of Biomedical Sciences, University of Copenhagen, Copenhagen N, Denmark. [3] These authors contributed equally: Ping Li, Kaituo Wang. ✉email: pontus.gourdon@med.lu.se

**P**-type ATPases form a large membrane protein superfamily that couples ATP hydrolysis to the transport of cargo across biological membranes[1]. They are divided into five classes, P1–5, based on sequence similarity and transport specificity[2]. While P1–3 represent ion pumps prevalent in all kingdoms of life, P4-ATPases are phospholipid flippases that do not occur in prokaryotes. Analogously, P5-ATPases are ubiquitously present in eukaryotic species only, representing the least characterized class physiologically, mechanistically, and structurally. They are subdivided into two subclasses, P5A and P5B, based on the conservation of residues in the transmembrane domain[3]. The yeast P5A-ATPase Spf1 (hereafter referred to Spf1) was recently determined to be a helix dislocase required for the removal of mis-inserted hydrophobic helices from the ER membrane[4]. P5B-ATPases are more divergent, with typically four members per species[5], including human ATP13A2–5. ATP13A2 resides in the lysosome[6], its malfunction leads to ER-associated cell death[7,8] and ATP13A2 mutations cause Kufor–Rakeb syndrome, a form of Parkinson's disease associated with dementia[9,10]. Equivalently, expression of ATP13A2 has been shown to protect against α-synuclein-induced cytotoxicity[11]. Recently, ATP13A2 was demonstrated to transport physiological polyamines such as spermine (SPM) from the lysosomal lumen into the cytosol[12]. Together, these results advocate a fundamental role of P5B-ATPases in lysosomal health, thus representing attractive targets for studies and novel treatments of neurodegenerative diseases.

Biochemical data suggest that P5B-ATPases comprise the conserved P-type ATPase core, with three cytoplasmic domains and a transmembrane domain similar to P2-, P3-, and P4-ATPases. However, P5-ATPases feature a unique N-terminus, with two additional TM helices, Ma and Mb, in P5A-ATPases, distantly reminiscent to the P1B-ATPases, which are predicted to be absent in P5Bs[5,13,14]. Thus, even the topology, overall architecture, and domain organization of P5Bs remain elusive. P5-ATPases are believed to exploit the classical P-type ATPase reaction cycle, with alternating access between inward-facing E1 and outward-facing E2 states, linked to autophosphorylation and dephosphorylation, respectively, through four principal states, E1–E1P–E2P–E2 (P denotes phosphorylated configurations) (Fig. 1a)[15,16]. Here, we provide a series of single-particle cryo-electron microscopy (cryo-EM) structures of the P5B-ATPase Ypk9 from the thermophilic yeast *Chaetomium thermophilum* (hereafter denoted Ypk9), granting new structural insights into transport and regulation mechanisms of P5B-ATPases.

## Results and discussion

**Overall structure and conformation.** Ypk9 was initially studied using nanodisc-reconstituted C-terminally green fluorescent protein (GFP)-fused sample. A cryo-EM structure, coined E2P[inhib], was determined at an average resolution of 3.5 Å, generated in the presence of the phosphate analog beryllium fluoride (BeF[3]−), which has been previously employed as a conformation-stabilizing tool for investigations of P-type ATPases[17] (Methods and Supplementary Table 1). The cryo-EM maps show well-resolved domains (Supplementary Figs. 1a and 2), enabling de novo model building of the entire ATPase, except for certain peripheral loops of the soluble domains, and some 200 N-terminal residues that are largely absent in the human P5B-ATPase members (Supplementary Fig. 3). The structure reveals a typical P-type ATPase fold, including the conserved cytosolic actuator (A-), phosphorylation (P-), and nucleotide-binding (N-) domains, and a transmembrane (M-) domain composed of ten transmembrane spanning helices, TM1–10 (Fig. 1). In addition, an N-terminal domain (NTD) structurally related to the one observed for the yeast P5A-ATPase Spf1 is detected[4] (Supplementary Fig. 4). Nevertheless, comparisons to the Spf1 structures reveal major differences, such as a unique NTD topology (see further below), and lack of the so-called arm domain, an α-helical elongation emerging from the P-domain of Spf1[4]. Furthermore, weak cryo-EM density is present immediately adjacent to the cytosolic leaflet, located between the first assigned residue of the NTD and the P-domain, though no secondary structure features are visible. The presence of an elongated N-terminus, prior to the NTD, appears unique to certain P5B-ATPases, as such extensions are absent in the P5A subclass (Supplementary Figs. 3 and 4). Structure-based alignments of the soluble A-, N-, and P-domains to different structures of the well-studied P2 sarco/endoplasmic reticulum Ca2+-ATPase (SERCA) and the P4-ATPase ATP11C-CDC50A[18] suggest that our structure resembles

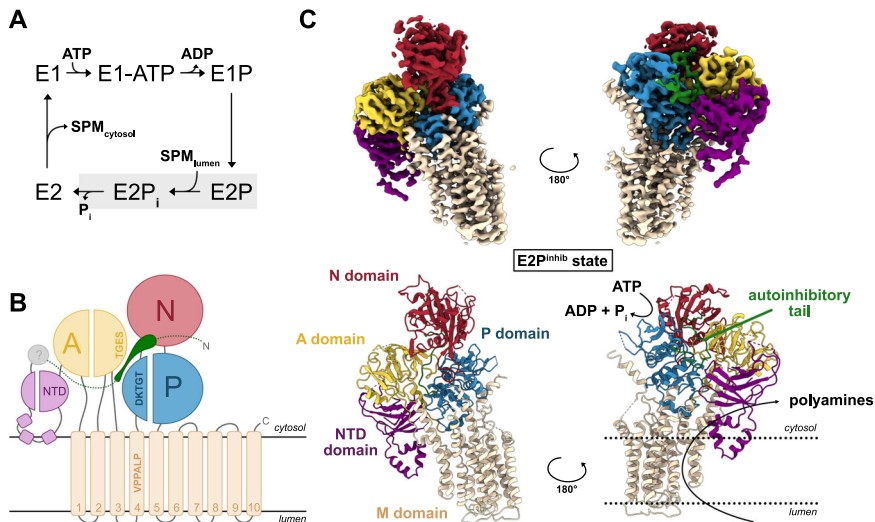

**Fig. 1 Overall P5B-ATPase structure and topology. a** The P5B-ATPase transport cycle likely follows the E1–E1P–E2P–E2 Post-Albers scheme shown here, with the part of the focus in this work highlighted in gray. ATP consumption energizes the transport of polyamines such as spermine (SPM). **b** The P5B-ATPase topology, with ten transmembrane helices TM1–10 and the cytosolic A-, P-, and N-domains are conserved among P-type ATPases. The P5-specific N-terminal domain (NTD, pink) harbors three short helices dipping into the membrane. Moreover, N-terminal residues form an auto-inhibitory tail (green) interacting with the A-, P-, and N-domains in the E2P[inhib] state. **c** EM density and cartoon representation of the E2P[inhib] conformation with contour level = 0.8.

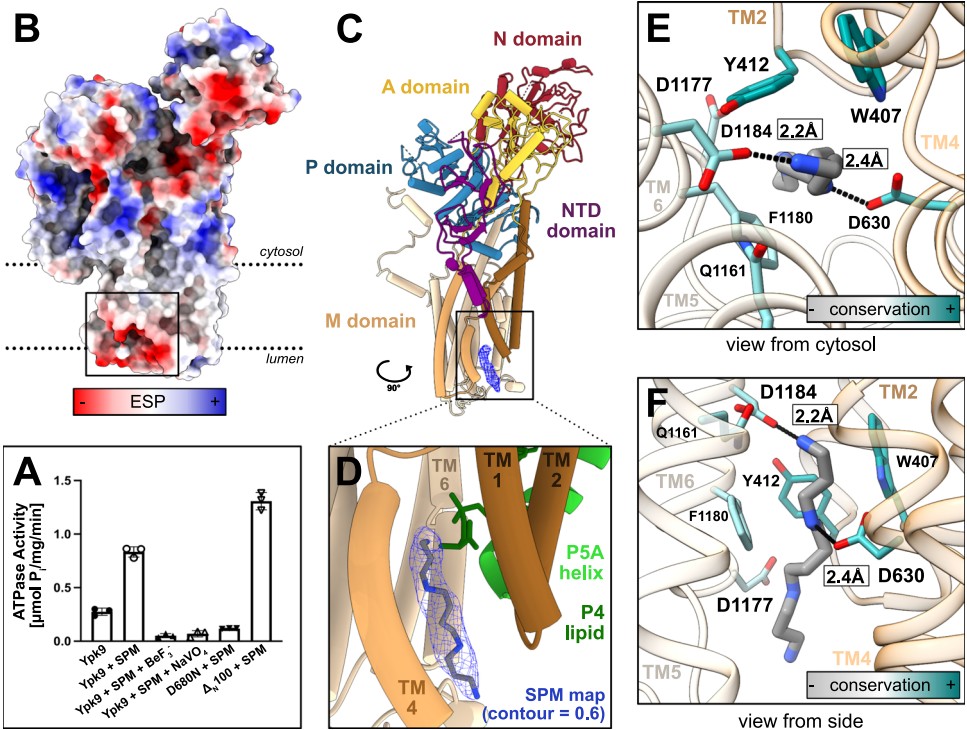

**Fig. 2 Polyamine uptake. a** ATPase activity of purified Ypk9 is stimulated by the polyamine spermine (SPM) and inhibited by the inhibitors $BeF_3^-$ and $NaVO_4^-$ ($n = 3$ independent measurements, error bars represent the mean with SD). The function is abolished by a classical P-type ATPase dead-mutation of the catalytic aspartate (D680N), and instead stimulated when the first 100 amino acids are removed ($\Delta_N$100). **b** Surface electrostatics of the E2.$P_i$ state. The luminal surface is highly electronegative, congruent with polyamine access. **c, d** Side-view and close-up of the SMP binding pocket. **d** Alignment of the E2.$P_i$ state and the cargo-bound structures of the P5A-ATPase Spf1 (PDB-ID 6XMU) and the P4-ATPase ATP8A1-CDC50 (6K7M), showing only their helix and lipid in green. SPM density is displayed as blue mesh, map contour = 0.6. **e, f** Close-view of the SPM binding pocket and coordinating residues. Side chains are colored by conservation, as determined by AL2CO implemented in ChimeraX[42,45], excluding side chains with poor conservation (<0.5).

an outward-facing E2P state, congruent with the open config-uration caught by $BeF_3^-$ in other P-type ATPases (Supplementary Fig. 5 and Supplementary Table 2). As an example, the conserved A-domain TGES dephosphorylation loop is located near the catalytic aspartate. Considering that the E2P state is known to be linked to counter cargo uptake from the outside in ion-transporting P-type ATPases, the obtained state is pre-sumably linked to polyamine recognition in P5B-ATPases, an enigmatic aspect of these proteins.

**Spermine uptake and transport.** The direct transport of the biogenic polyamines SPM and spermidine (SPD) from the lyso-some to the cell interior by the human P5B-ATPase ATP13A2 has recently been demonstrated[12]. To unravel the underlying molecular transport mechanism, we first confirmed SPM-stimulated activity for Ypk9 in vitro, using a dead-mutant of the catalytical phosphorylation site, D680N, as a control (Fig. 2a). Next, additional structures were determined, all in the presence of SPM, and using N- or C-terminally linked GFP protein. We exploited conditions in the presence again of $BeF_3^-$ (yielding a structure at 3.7 Å resolution overall, denoted E2P*), or without a similar compound (3.5 Å, E2.$P_i^{SPM}$) (Supplementary Table 1, Supplementary Figs. 1a, b and 2). Moreover, the alternative phosphate mimic aluminum fluoride, $AlF_4^-$, was applied (3.4 Å, E2.$P_i^{AlF/SPM}$) (Supplementary Figs. 1b and 2), representing a compound known to catch a transition state of depho-sphorylation in-between E2P and E2, E2.$P_i$, which is cargo-occluded in ion-transporting P-type ATPases[19]. Overall, the dif-ferent SPM data collected show well-resolved soluble domains, although the N-domain density is somewhat poor (Supplementary Fig. 1b). Structural analyses to SERCA and

ATP11C-CDC50A suggest the SPM-structures are indeed caught in separate E2P and E2.$P_i$ configurations, with the E2.$P_i^{SPM}$ structure indicating the presence of cargo is sufficient to stabilize an E2.$P_i$ state (Supplementary Table 2). While the E2.$P_i$ structures are highly similar, distinctions are identified between the two recovered E2P states (Supplementary Fig. 6, see further below).

Interestingly, the E2.$P_i$ models exhibit a highly electronegative luminal surface (Fig. 2b and Supplementary Fig. 7), from which an opening emerges between TM2, 4, and 6, reaching the central TM4 kink of all P-type ATPases, and the VPPALP P5B-signature motif. The cleft harbors an additional, elongated density compatible with the presence of SPM (Fig. 2c, d). The pocket is permitted by an unwound TM4 and is formed by several electronegative and hydrophobic side chains, of which some, i.e., the P5B-conserved D630 in TM4, Y1157 in TM5, D1177, and D1184 in TM6, appear directly interacting with SPM (Fig. 2e, f, Supplementary Figs. 3 and 8). Indeed, in vitro studies on the D962N variant of ATP13A2 (D1184N in Ypk9) revealed a complete loss of ATPase activity and spermine-induced dephosphorylation[12], highlighting the central role of this residue in polyamine detection and transport. As such, the composition of the cavity also allows binding of shorter physiological polyamines, e.g., SPD and putrescine, in a similar manner. Moreover, we note that the TM2, 4, and 6 cargo-binding area overlaps well with the one observed in both P5A-ATPases, Spf1 (PDB-ID 6XMU), and P4-ATPases, ATP8A1-CDC50a (6K7M) and ATP11C-CDC50a (6LKN and 7BSU) (Fig. 2d)[4,18,20,21], and that these separate cargoes are less occluded as compared to P2- and P3-ATPases. Thus, it appears that critical features of the transport pathway are conserved independent of lipid, helix, or

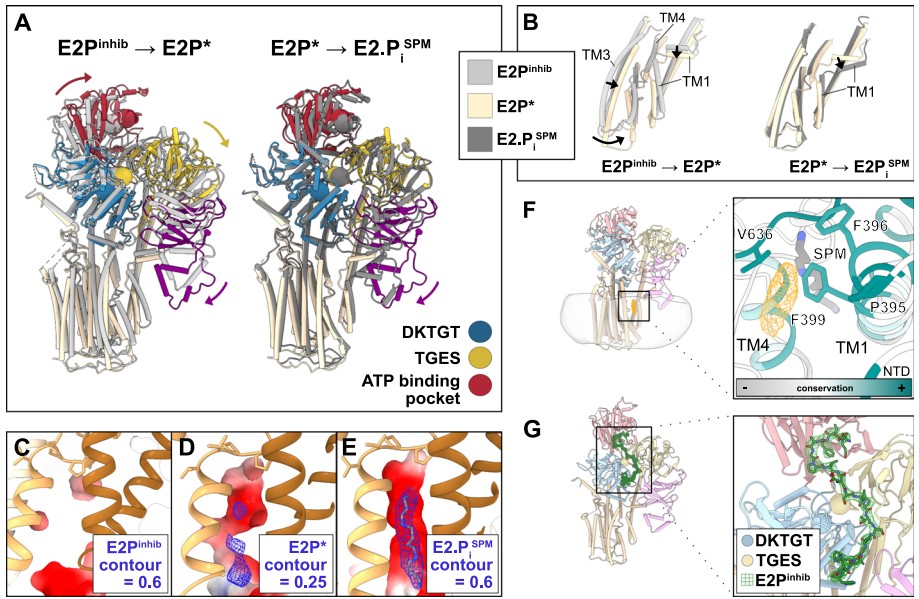

**Fig. 3 Conformational changes during polyamine uptake, lipid regulation, and auto-inhibition. a** Alignment of the generated structures on the P domain. Conserved motifs in the soluble domains are shown as spheres. **b** Alignment on TM5–10, and close-view of conformational changes in TM1–6. **c–e** Close-view of the polyamine binding pocket in the E2P$^{inhib}$ (**c**), E2P* (**d**), and E2.P$_i$ (**e**) states. EM density around SPM is shown in the blue mesh. **f** Side-view and a close-up of the lipid density in a conserved cleft formed by TM1, 3, and 4, illustrating the position of the lipid at the cytosolic membrane leaflet. The close-view shows conserved side chains in proximity to the lipid density, colored by conservation, as determined by AL2CO implemented in ChimeraX[42,45], excluding side chains with poor conservation (<1). **g** Side- and close-views showing the location of the auto-inhibitory N-terminal residues (mesh, dark green) in the E2P$^{inhib}$ state. The DKTGT and TGES motifs of the P- and A-domains are shown as spheres. The autoinhibitory residues are interacting with the TGES loop and the nucleotide-binding pocket of the N domain.

polyamine cargo, in agreement with the shared common evolutionary origin of P4- and P5-ATPases[5]. Nevertheless, details of the mechanistic principles differ in-between the subclasses, i.e., SPM appears more protein-buried than the helix of P5A-ATPases, and yet solvent-exposed to the luminal side, similar to the lipid headgroups in BeF$_3^-$-stabilized conformations of P4-ATPases[18,20].

Comparison of the structures reveals major rearrangements of the TM helices linked to SPM uptake (Fig. 3a, b; Supplementary Figs. 7 and 8). The electronegative luminal surface is present in all structures (Supplementary Fig. 7). However, while we do not detect polyamine indications in the E2P$^{inhib}$ structure, poor cryo-EM density for SPM, somewhat shifted toward the luminal side compared to the E2.P$_i$ structures are available for E2P* (Fig. 3c–e). The SPM binding pocket is not yet defined in the E2P$^{inhib}$ conformation, where the luminal parts of TM1–2, TM3–4, and TM5–10 are further apart. Based on these observations, we speculate that the affinity for SPM increases from the E2P$^{inhib}$ to the E2P* and E2.P$_i$ states. The structural changes that orchestrate the SPM pocket are coupled to a rotation of the A-domain, leading to a 6 Å movement of the TGES dephosphorylation loop as observed in other P-type ATPases, and a similar movement of the adjacent NTD (Fig. 3a).

**A Ma–Mb depleted NTD.** P5A-ATPases feature two extra TM helices, Ma and Mb, emerging as an insert to the NTD. However, previous biochemical studies on P5B-ATPases have indicated a topology with ten transmembrane helices only, and yet NTD-linked lipid-regulation[5,14], leaving the assembly and mechanistic role of the N-terminus of P5A-ATPases elusive. While peripheral parts are somewhat discontinuous in the densities, our structures clearly demonstrate that Ma and Mb are absent in P5B-ATPases (Fig. 1b, c). Instead, an amphipathic loop is formed by three α-helices likely embedded in the membrane interface. Several of the

residues in this stretch are highly preserved, including the hydrophobic L246 and W250, which may serve as a lipid bilayer anchor, as well as helix-terminating G242 (Supplementary Fig. 3). However, the remaining part of the NTD, essentially composed of a seven-stranded β-barrel directly linked to the A-domain, superposes well with the P5A structure (Supplementary Fig. 4), suggesting a common origin of the NTD for all P5-ATPases.

What is then the functional role of the NTD? Notably, in the membrane of the E2.P$_i$ structures, a single strong additional density not explained by the protein or SPM is located in a cleft formed by transmembrane helices TM1, 3, and 4 (Fig. 3f). The feature is in close proximity (<6 Å) to the conserved V636 of the VPPALP signature motif of TM4, but also the membrane-embedded loop of the NTD. Previous studies have demonstrated a direct interaction of both phosphatidic acid (PA) and phosphatidylinositol(3,5)bisphosphate (PI(3,5)P2) with three separate stretches of the ATP13A2 NTD and a stimulatory effect on ATPase activity[14]. Interestingly, two of these NTD regions are located in the membrane-dipping region of the Ypk9 NTD (L248–P252 and K257–K262), and the remaining portion is in close proximity to the observed density. We, therefore, hypothesize that the observed density relates to a lipid of the cytosolic membrane leaflet. The lipid may serve a regulatory function, linking the NTD and A-domain to TM1, 3, and 4. However, the presence of an unpredicted glycosylation site or even a non-native detergent molecule cannot be excluded (alternative lipid effects via the highly electropositive region around the C-terminus are also possible, Supplementary Fig. 9). The location close to the SPM-interacting 'spine' M4 transmembrane helix of P-type ATPases may permit coupled binding of the lipid and cargo in P5B-ATPases. Indeed, the NTD-domain is displaced between the E2P and E2.P$_i$ structures (Fig. 3a). We also note that the topological differences of P5A- and P5B-ATPases in this way may represent two separate principles of conferring lipid-anchored NTD-influence to the ATPase core.

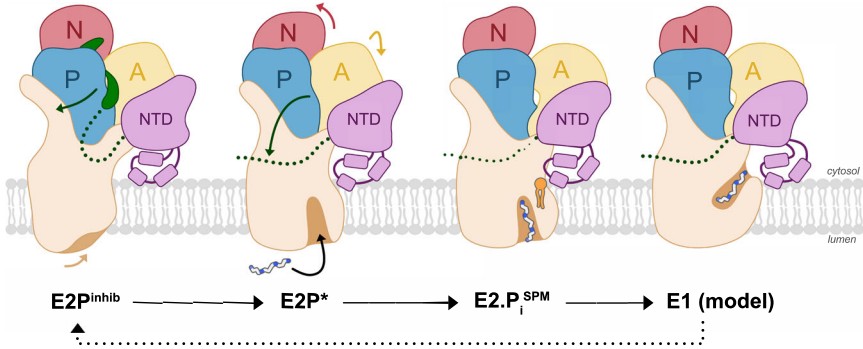

**Fig. 4 Proposed transport mechanism.** We suggest P5B-ATPases rest in a phosphorylated state, auto-inhibited through interactions of N-terminal residues with the cytosolic A-, P-, and N-domains (E2P$^{inhib}$). Polyamine binding and the coupled dephosphorylation lead to the release of the auto-inhibitory loop, and the formation of a binding pocket between TM 2, 4, and 6 (E2P*). It is possible that the formation of the cavity and polyamine uptake may be stimulated through interactions of the NTD with a regulatory lipid (E2.P$_i^{SPM}$, orange). Dephosphorylation and the subsequent E2 → E1 transition will then lead to cargo release to the cytosol, priming the ATPase for nucleotide binding, phosphorylation, and a new transport cycle.

**Auto-inhibition**. The E2P structures exhibit another peculiar feature. Sandwiched in-between the soluble domains, a long continuous density is observed. While poorly resolved in the E2P* data, we assign the stretch to residues D74-G97 of the Ypk9 N-terminus based on visible side chains in the E2P$^{inhib}$ cryo-EM density (Fig. 3g). Strikingly, it interacts with several key aspects for ATP-hydrolysis and auto-dephosphorylation which are omnipresent among P-type ATPases, namely the TGES dephosphorylation motif of the A-domain, the nucleotide-binding pocket in the N-domain, and parts of the surface of the P-domain[22]. The C-terminus of the loop is also in close proximity to the above-mentioned poorly resolved density that likely bridges to the NTD (Supplementary Fig. 10).

While such properties have not been assigned to P5-ATPases previously to our knowledge, this observation brings to mind the regulatory termini known to be present in many P-type ATPase subclasses, including P1B, P2B, P3A, and P4-ATPases[23–26]. In the more P5-alike P4 class, auto-inhibitory C-terminal domains have been detected in both human ATP8A1–CDC50a and yeast Drs2p–Cdc50p, tightly blocking the nucleotide-binding pocket[18,27]. As the loop was exclusively observed in a BeF$_3^-$-stabilized E2P conformation, it was suggested to specifically stabilize this particular state of P4-ATPases[18]. A comparison of the N-terminal loop in Ypk9 with the C-terminal regulatory domain of P4-ATPases exposes distinct similarities, such as the interference with the nucleotide-binding pocket and binding of the stretch in the E2P but not the E2.P$_i$ states (Supplementary Fig. 10). Indeed, a Ypk9 form lacking the first 100 residues, Ypk9Δ$_N$100, also demonstrates higher activity in our in vitro assay (Fig. 2a). Thus, it appears as if the observed N-terminal feature indeed serves an auto-inhibitory role, perhaps exclusively stabilizing the E2P conformation, and that the P4- and P5B-ATPase regulatory mechanisms share common features. The aforementioned notion is not in disagreement with the observed "resting" autophosphorylated configuration detected in ATP13A2[14]. This is because the N-termini of P5B-ATPases are not conserved (Supplementary Figs. 3 and 4), and such a mechanism would likely be limited to P5B-ATPases with an N-terminus of sufficient length, e.g., Ypk9. Also, complementary biochemical data are needed to confirm the proposed autoinhibition mechanism.

Nevertheless, these inhibitory P4- and P5B-loops also display unique aspects, such as that they emerge from different termini. In this light, the configuration of the C-terminus, following the last transmembrane helix, TM10, is of interest. A cytosolic extension is indeed present also in Ypk9, and while the preceding membrane-spanning part superposes well with other P-type

ATPases, the soluble portion remains helical but is kinked near the conserved E1351 (Supplementary Fig. 9). Next, the Ypk9 C-terminus forms an additional small, soluble helix, which directly interacts with the P-domain, compatible with regulatory properties as it may stabilize the relative organization of the P- and TM-domains. However, the localization of the C-terminus is maintained in all our structures, including constructs with N- and C-terminal GFP-fusions, hence excluding conformational bias caused by construct design, rendering it difficult to unequivocally assign a critical functional role also to the C-terminus of P5B-ATPases.

**Proposed transport mechanism**. P5B-ATPases have previously been suggested to follow the conserved Post-Albers reaction cycle of P-type ATPases[15,16]. Presumably, E1 to E1P autophosphorylation of the catalytic aspartate occurs in a polyamine-independent manner[14]. Although this contrasts to ion-transporting P1–3-type ATPases, this is similar to P4-ATPases[26], again highlighting the common evolutionary origin of these subfamilies. In the E2P$^{inhib}$ structure following E1P, the N-terminus is likely auto-inhibiting the pump through interference with the soluble A-, P-, and N-domains, thereby preventing turn-over (Fig. 4).

Formation of the cargo-binding pocket due to shifts of TM3 and 4 is likely via TM1, 2, and 3 coupled to major rearrangements of the soluble domains, in particular displacements of the A- and NTD-domains. As observed in our E2P* structure, these shifts weaken the interactions between the N-terminal loop and the ATPase core, leading to dissociation of the auto-inhibitory stretch from the soluble A-, P-, and N-domains. Simultaneously, indications of SPM between TM2, 4, and 6 are detected, and we denote this state as E2P*, as dephosphorylation has been initiated. The structural rearrangements may be reinforced through binding of regulatory lipids, possibly between TM1, 3, and 4 (as observed in our E2.P$_i^{SPM}$ structure), which is sensed by the NTD, thereby facilitating rotation of the NTD/A-domain. This agrees with the previously reported stimulatory effect of PA and PI(3,5)P2 on ATP13A2 turnover, which was abolished through mutations in the membrane-dipping loop of the NTD[14]. However, the release of the auto-inhibitory loop could as well be achieved through interactions with regulatory lipids and/or auxiliary proteins, as observed for the yeast P4-ATPase Drs2p[28,29], or phosphorylation, as for human ATP8A2[30]. Equivalently, completion of cargo-binding is linked to the proceeding E2.P$_i$ state, where SPM is accommodated in a cavity on the luminal side, coordinated by conserved acidic and hydrophobic residues, including the conserved VPPALP motif.

Cargo transport and release to the cytosol is then likely accomplished following dephosphorylation, in the E2 to E1 transition. We propose the exit pathway is formed by M2, 4, and 6, similar to the mechanism assumed for P4- and P5A-ATPases[4,18,20]. The absence of conserved and/or electronegative residues in this cleft suggests a direct release of the cargo from the observed high-affinity SPM binding site to the cytosol. Alternatively, due to the conserved nature of TM1 among P5B-ATPases, the exit pathway may be composed of TM1, 3, and 4, similar to the ion-uptake pathway in P1–3-type ATPases. This would, however, require a rather complex release mechanism, including re-arrangement of TM4 relative to TM1 to permit the polyamine to reach TM1, and we, therefore, believe that TM1 fulfills a role in lipid regulation rather than in cargo release.

The structures of Ypk9 presented here provide novel insights into the overall architecture and polyamine transport mechanism of P5B-ATPases. Ypk9 exhibits ten TM helices and a soluble N-terminal domain (NTD) in close proximity to the membrane, perhaps involved in lipid regulation. An N-terminal autoinhibitory loop likely stabilizes the E2P state, regulating polyamine turnover. Spermine uptake occurs from the luminal side, through a conserved electronegative pocket formed by TM2, 4, and 6. It is expected that conformational changes upon dephosphorylation trigger cargo translocation to the cytosol. However, additional structural and functional characterization is essential to fully elucidate the P5B-ATPase transport mechanism.

## Methods

**Cloning, overproduction, and purification of Ypk9.** Full-length Ypk9 (Uniprot G0S7G9) gene was amplified using polymerase chain reaction with Phusion™ High-Fidelity DNA Polymerase (ThermoFisher scientific) from genomic DNA of *C. thermophilum* (DSM1495) using the primers ctP5B-22b-F, ctP5B-22b-R (Supplementary Table 3), and then cloned into the pET-22b vector using the NEBuilder HiFi DNA Assembly Master Mix. The seven introns were individually removed using Δintron1-F/Δintron1-R, Δintron2-F/Δintron2-R, Δintron1-2/Δintron2-R, Δintron3-F/Δintron3-R, Δintron4-F/Δintron4-R, Δintron5-F/Δintron5-R, Δintron6-F/Δintron6-R, and Δintron7-F/Δintron7-R primer pairs listed in Supplementary Table 3. The ctP5B-N-GFP-F/ctP5B-N-GFP-R and ctP5B-C-GFP-F/ctP5B-C-GFP-R primer pairs were used for fusion with N- and C-terminal GFP, respectively. The ΔN100-F/ctP5B-C-GFP-R primer pair was used for generating the ΔN100 deletion form. The gene was confirmed by sequencing using Macrogen Europe. A TEV cleavage site, ENLYFQ, GGGGS linker, and green fluorescence protein (GFP) with a 10xHis tag were fused to the N- (E2P* and E2.P$_i$$^{ALF/SPM}$ structures) or C-terminus (E2P$^{inhib}$ and E2.P$_i$$^{SPM}$) of Ypk9, and cloned into the pEMBLyex4 expression vector[31]. The expression plasmids pEMBLyex4-Ypk9-TEV-G4S-GFP-His$_{10}$ and pEMBLyex4-10xHis$_{10}$-GFP-G4S-TEV-Ypk9 were separately transformed into the PAP1500 *Saccharomyces cerevisiae* strain[31] using the LiAc/SS carrier DNA/PEG method[32] and plated on SD agar plates (1.9 g/L Yeast Nitrogen Base, 5 g/L Ammonium sulfate, 20 g/L Glucose, 30 mg/L Lys, 60 mg/L Leu, 1.5% (w/v) Agar).

Protein production was accomplished according to previously reported protocols[31] with small modifications. Briefly, single colonies from freshly transformed plates were inoculated in 5 mL synthetic minimal (SD) medium supplemented with leucine (Leu) (60 mg/L) and lysine (Lys) (30 mg/L) at 30 °C with 220 rpm shaking for 24 h. 2 mL overnight preculture was diluted in 100 mL SD media (1.9 g/L yeast nitrogen base, 5 g/L ammonium sulfate, 20 g/L glucose) supplied with 30 mg/mL Lys and cultivated at 30 °C with 200 rpm shaking for around 30 h. The cells were pelleted and transferred to 800 mL expression media (1.9 g/L yeast nitrogen base, 5 g/L ammonium sulfate, 5 g/L glucose, 1.1 g/L -Ile,-Ura dropout amino acid mixture, 3% (v/v) glycerol) at 30 °C for 20 h. Protein production was induced through the addition of 2% (w/v) galactose dissolved in expression media lacking glucose, and protein production was conducted for 24 h at 25 °C.

The cells were washed, resuspended in lysis buffer (20 mM Tris-HCl pH = 7.5, 100 mM NaCl, 5% glycerol, 2 mM BME, 5 mM EDTA), and disrupted by high-pressure homogenization (Xpress). Next, the broken cells were resuspended in lysis buffer at 50 mg/mL. Unbroken cells and cell debris were removed by centrifugation at 4000×*g* for 10 min and crude membranes were isolated by 90 min centrifugation at 165,000×*g*. The membrane pellets were solubilized using solubilization buffer (50 mM Tris-HCl pH = 7.5, 150 mM NaCl, 20% (v/v) glycerol, 2 mM BME) supplied with 2% (w/v) *N*-dodecyl β-d-maltoside (DDM), 0.2% (w/v) cholesterol hemisuccinate (CHS) and EDTA-free protease inhibitor cocktail to a final concentration of 50 mg/mL for 2 h. Insolubilized material was removed through 30 min centrifugation at 190,000×*g*. The supernatant was diluted 5 times in buffer

A (25 mM Tris-HCl pH = 7.5, 150 mM NaCl, 10% (v/v) glycerol, 2 mM BME) and loaded on a prepacked Sepharose 5 mL Histrap column equilibrated in 30 mL buffer A containing 0.05% (w/v) DDM and 0.005% (w/v) CHS. Three washing steps were employed to remove unspecifically bound proteins, i.e., 30 mL of buffer A with 10 mM (final) imidazole, 0.1% (w/v) DDM and 0.01% (w/v) CHS; then 20 mM imidazole, 0.075% (w/v) DDM and 0.0075% (w/v) CHS; and finally 40 mM imidazole, 0.05% (w/v) DDM and 0.005% (w/v) CHS. Bound protein was eluted in buffer A with 0.05% (w/v) DDM and 0.005% (w/v) CHS and 300 mM imidazole, using a one-step procedure. The eluate was assessed by sodium dodecyl sulfate–polyacrylamide gel electrophoresis (SDS-PAGE) and the fractions containing Ypk9 pooled and concentrated using Ultra-15 centrifugal concentrators (Amicon) with 100 kDa MW cut-off. The sample was then applied to a Superose6 size exclusion chromatography column (Cytiva) equilibrated in buffer A with 0.05% (w/v) DDM and 0.005% (w/v) CHS. The peak fractions were pooled and concentrated to 5 mg/mL for the Nanodisc reconstitution or stored at −80 °C for further application. The protein purity was assessed using SDS-PAGE.

**MSP1E3D1 production and purification.** The MSP1E3D1[33] plasmid was transformed into BL21(DE3) cells. Single colonies were incubated in 10 mL LB medium supplied with 25 mg/L kanamycin and cultivated 16 h at 37 °C with shaking 200 rpm. The pre-culture was transferred into 1 L LB medium with 25 mg/L kanamycin and incubated for 3 h at 37 °C with shaking at 200 rpm. Protein production was induced through the addition of 1 mM IPTG at an OD600 of 0.6, and then continued at 37 °C for 1 h following 3 h at 30 °C. The cell pellets were washed with PBS buffer and then re-suspended in PBS buffer supplied with 2 mM MgCl$_2$, 0.01 mg/mL DNaseI and EDTA-free protease inhibitor cocktail (50 mg/mL). The cells were disrupted by 20 min sonication. Unbroken cells and cell debris were removed by centrifugation at 75,000×*g* for 1 h. The supernatant was filtered (0.45 μm) and applied to Ni-NTA resin (Cytiva) for affinity purification. The column was then washed with 100 mL of the following buffers: (1) 40 mM Tris-HCl pH = 8.0, 300 mM NaCl and 1% (v/v) Triton; 2) 40 mM Tris-HCl pH = 8.0, 300 mM NaCl, 50 mM Na-Cholate and 20 mM Imidazole; 3) 100 ml 40 mM Tris-HCl pH = 8.0, 300 mM NaCl and 40 mM Imidazole. The protein was eluted in 15 mL elution buffer (40 mM Tris-HCl pH = 8.0, 300 mM NaCl, and 400 mM Imidazole). TEV protease was added to cleave the His-tag, and the sample was dialyzed for 16 h against 20 mM Tris-HCl pH = 7.4, 100 mM NaCl and 0.5 mM EDTA. To separate the cleaved target protein from TEV protease, the sample was again applied to Ni-NTA resin, the flow-through collected and concentrated to 5 mg/mL, flash-frozen in liquid nitrogen, and stored at −80 °C until further usage.

**Nanodisc reconstitution.** Yeast polar lipids dissolved in chloroform (Avanti Lipids, 25 mg/mL) were gently dried under nitrogen gas and dissolved in 20 mM Tris-HCl pH = 7.5, 100 mM NaCl, 0.5% (w/v) DDM. The E2.P$_i$$^{SPM}$ structure was determined using protein reconstituted into MSP1D1 with a molar ratio of Ypk9:MSP1D1:lipids of 1:4:100. All the other structures were generated in the presence of MSP1E3D1, reconstituted with a molar ratio of Ypk9:MSP1D1:lipids of 1:10:300. The lipids were added to the protein sample, gently mixed, and incubated for 30 min, and then the MSP protein and buffer added. 200 mg bio-beads SM2 (Bio-Rad) were supplemented, and the sample was incubated for 2 h to remove detergents. The protein nanodisc complex was further purified by IMAC and size-exclusion- chromatography using a Superdex 200 column (Cytiva). The Ypk9 nanodisc peak fractions were pooled and concentrated to 1 mg/mL for the cryo-EM sample preparation.

**Cryo-EM sample preparation.** Purified Ypk9 in nanodiscs was frozen at a concentration of 0.8–1.2 mg/mL. Quantifoil 1.2/1.3 holy carbon grids were glow-discharged using a Leica Coater ACE 200 for 40 s with 10 mA current. The grids were prepared using a Vitrobot Mark IV operated at 100% humidity and 4 °C. In total, 3 μL of purified protein was applied to each grid, incubated for 5 s, blotted for 3 s, and then plunge frozen into liquid ethane. Frozen grids were stored in liquid nitrogen until data collection. The different protein conformations were stabilized using supplements added immediately before freezing: E2P$^{inhib}$ (1 mM BeF$_3$$^-$); E2P* (1 mM SPM and 1 mM BeF$_3$$^-$); E2.P$_i$$^{ALF/SPM}$ (1 mM SPM and 1 mM AlF); E2.P$_i$$^{SPM}$ (1 mM SPM and 1 mM MgCl$_2$). All SPM containing samples were first incubated for 1 h at 18 °C and then heat-activated in a 40 °C water bath for 5 min immediately before freezing. The E2P$^{inhib}$ sample was only incubated at 18 °C without heat treatment.

**Cryo-EM data collection and data processing.** The cryo-EM datasets were collected on two separate Titan Krios electron microscopes (FEI) operated at 300 kV with either a Falcon3 detector in counting mode or a Gatan K3 detector in super-resolution mode. For the Falcon3 dataset, the pixel size was set to 0.832 Å and the total dose was 40 e/Å2 in 40 frames. For the Gatan K3 datasets, the pixel size was 0.54295 Å and the total dose was 50 e/Å2 in 40 frames. An energy filter at 20 eV was applied for the Gatan K3 data collection. All data were processed using cryosparc[34] following the procedures outlined in Supplementary Figs. 12–15 and Supplementary Table 1. For the Falcon3 dataset, the data were initially processed using full-frame motion correction and patch CTF determination. Blob particles with a diameter of 80–120 Å were picked without templates and extracted to a box

size of 360 pixels (300 Å diameter) using local motion correction with dose-weighting. Extracted particles were subjected to several rounds of reference-free two-dimensional class averaging to remove obvious junk, contamination, and empty nanodiscs. The cleaned particle set was processed following standard cryosparc workflow steps including ab-initial model reconstitution, multiple rounds of heterogeneous refinement, and non-uniform refinement iterations[35]. For the Gatan K3 dataset, the data were first motion-corrected using patch motion correction with 2× binning to micrographs with a pixel size of 1.086 Å. CTF parameters were determined using patch CTF. Blob particles were picked and extracted to a box size of 288 pixels (312 Å diameter). After similar data processing steps as described for the Falcon3 datasets, the last particle set was re-extracted using local motion correction to super-resolution with a box size of 600 pixels to push the resolution of the final map.

**Model building**. The initial models were generated by the SWISS-MODEL online server[36] using the corresponding sequences of Ypk9 and the EM structures of Spf1 (PDB-ID 6XMP and 6XMT) as templates. The missing residues were modeled as unstructured loops. The full-length models were then fitted into the corresponding cryo-EM density maps using the molecular dynamics flexible fitting (MDFF) method in an implicit solvent with secondary structure, cis-peptide, and chirality restraints to prevent overfitting[37]. The E2P$^{inhib}$ model was then further refined by multiple rounds of manual adjustments in Coot[38] and using MDFF and phenix_real_space_refine[39]. The dipping region of the NTD domain was built manually in Coot guided by side chain features in the density. The autoinhibitory tail was assigned manually based on density features. The other structures were generated using the same strategy, based on the first-built E2P$^{inhib}$ structure. MolProbity implemented in Phenix was used for model validation[40].

**Figure preparation**. All figures were generated using UCSF Chimera[41] and ChimeraX[42]. Alignments were performed using Clustal Omega[43] and visualized using ESPript 3.0[44].

**ATPase activity**. ATP turnover stimulated release of inorganic phosphate was performed using a commercial ATPase/GTPase Activity Assay Kit (MAK113 from Sigma). Absorbance was measured at 620 nm detecting liberated phosphate colored by malachite green. 40 ng purified Ypk9 in DDM was used for 50 μL final reaction mixture with assay buffer (50 mM MOPS-KOH pH = 7.0, 100 mM KCl, 5 mM MgCl$_2$, 2 mM 1,4-Dithiothreitol, 0.05% (w/v) DDM, 0.005% (w/v) CHS). If present, the samples also contained final concentrations with 6 mM SPM, 1 mM BeF$_3^-$ and 1 mM NaVO$_4$. The reactions were initiated through supplementation of 1 mM ATP and performed at 37 °C for 20 min. Next, 20 μL of the samples were transferred to a 384-well microplate, mixed with 80 μL reagent, and incubated for an additional 15 min at 18 °C. The measurements of absorbance at 620 nm were carried out using a Multiskan Sky Microplate Spectrophotometer (Thermo Scientific). The data in Fig. 2A are shown as three independent measurements based on one purification for each Ypk9 form, and the error bars represent the mean with SD. The analysis of the SPM transport assay was performed using Graph Pad Prism 9.

**Reporting summary**. Further information on research design is available in the Nature Research Reporting Summary linked to this article.

## Data availability

All data and materials supporting the findings in the paper are available from the corresponding author upon reasonable request. The structural coordinates and EM data have been deposited in the Protein Data Bank and in the Electron Microscopy Data Bank with the following accession numbers: E2P$^{inhib}$ (P8, E4), E2P* (P5, E3), E2.P$_i^{AlF/SPM}$ (P1, E1), and E2.P$_i$SPM (P3, E2). Source data are provided with this paper.

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

## Acknowledgements

We would like to thank Julian Conrad and Marta Carroni at SciLifeLab Stockholm as well as Tillmann Hanns Pape at CFIM University of Copenhagen for assistance with the Cryo-EM data collection. We acknowledge access to the computational resources from the Danish National Supercomputer for Life Sciences (Computerome). CG is currently paid by The BRIDGE—Translational Excellence Program at the University of Copenhagen funded by the Novo Nordisk Foundation (NNF18SA0034956). KW is supported by the Lundbeck Foundation. PG is supported by the following Foundations: Lundbeck, Knut and Alice Wallenberg, Carlsberg, Novo-Nordisk, Brødrene Hartmann, Agnes og Poul Friis, Augustinus, Crafoord as well as The Per-Eric and Ulla Schyberg. Funding is also obtained from The Independent Research Fund Denmark, the Swedish Research Council, and through a Michaelsen scholarship. The funders had no role in study design, data collection, and analysis, decision to publish, or preparation of the paper.

## Author contributions

P.L. performed cloning, overproduction, purification, and nanodisc-reconstitution for the structural and functional studies. Cryo-EM sample preparation and data analysis were primarily performed by K.W.. P.L. and K.W. assisted with Cryo-EM data collections at SciLifeLab Stockholm, and K.W. executed the data collection at CFIM University of Copenhagen. P.L., N.S., C.G., and K.W. built and refined the structures. N.S. prepared the figures except for Supplementary Figs. 12–15 that were generated by K.W. P.L., K.W., and P.G. contributed to the identification of the scientific problem and experimental planning. P.L., N.S., C.G., K.W., and P.G. conducted data analysis and interpretation. N.S. and P.G. wrote the first draft. All authors commented on the paper. P.G. supervised the project.

## Funding

## Competing interests

The authors declare no competing interests.
