## [Peer Review File · Nature Communications]

Reviewers' Comments:

Reviewer #1:

Remarks to the Author:

What are the noteworthy results?

This paper describes four atomic resolution structures obtained by CryoEM of γ PK9 from the yeast *C. thermophilus*, which is related to the human ATP13A2 protein associated with juvenile Parkinson's, and other neurological pathologies. P5B-ATPases have recently reported to transport spermidine and other polyamines from the lysosome to the cytosol. This transport specificity, unique among P-ATPases, correlates with structural features not conserved in other members of this superfamily.

- Will the work be of significance to the field and related fields? How does it compare to the established literature? If the work is not original, please provide relevant references.

The data presented in this manuscript is highly relevant since it is the first report of an atomic model structure of a P5B-ATPase. Based on the observations the authors propose several functional mechanisms that are worth to consider and will open the door to future experimentation.

- Does the work support the conclusions and claims, or is additional evidence needed?

The manuscript has structural information highly valuable in itself. The authors engage in several interpretations which need additional experimental evidence.

- Are there any flaws in the data analysis, interpretation and conclusions? - Do these prohibit publication or require revision?

I find no flaws in the data or interpretations although some of the data included show a poor fit to the EM map. I think the manuscript is a valuable contribution but may be improved by a Thoughtful revision.

- Is the methodology sound? Does the work meet the expected standards in your field?

The methodology is sound. One thing that surprises me is that according to the PDB validation a great number of residues (54% for the AF3/SPM structure) exhibit a poor fit to the EM map. This value is substantially lower for other CryoEM structures of P-ATPases that have been reported previously and somehow it contrasts with the reported 3.7 Å resolution.

- Is there enough detail provided in the methods for the work to be reproduced?

yes.

Major points.

1. Manuscript organization. The manuscript is organized in Abstract, Introduction, Results and Summary. As written the Results experimental observations are mixed with interesting but speculative interpretations without experimental support. I suggest the authors to make separate sections of Results with a detailed description of each new structure and a Discussion section containing the functional interpretations and possible implications of the new structural features.

2. page 3 line 11 "In addition, an N-terminal domain (NTD) related to the one observed for the P5A-

ATPases is detected". Which is related NTD sequence of P5A and P5B? I don't see there is much sequence conservation in this region.

3. page 3 line 18 "Consequently, the N-terminus prior to the NTD appears unique to the P5B-ATPases...." The authors define NTD as starting at CtPK9 residue S206 forward. What is the region prior to NTD the authors are referring to? ATP13A2 has only 29 residues. CtSpf1 has 18 residues. It seems to me that only the yeast yPK9 have a long segment previous to NTD.

4. page 5 line 5 "This observation is congruent with polyamine binding incapacity for the E2P inhib state determined in the absence of SPM...." I believe this is an interesting interpretation that I would move to a Discussion section. There is no experimental data substantiating the lack of SPM binding to E2Pinhb.

5. page 5 .Second paragraph. The authors describe the structure and speculate about the possible function of NTD. It would help the reader if NTD would be better defined including a new figure showing exclusively the sequence and structural alignments of NTD of P5B and P5A (this will also help in Mayor point 2).

6. page5 .Third paragraph. The authors detect an unexpected density in E2Pi structure. Then the authors go on assuming that this density "relates to a lipid of the cytosolic membrane leaflet". This section seems interesting but speculative in the absence of a better resolution. Experimental data indicating that the detected density is compatible with phosphatidic acid or PI(3,5)P2 needs to be provided. In addition to add support of the authors claim, during the nanodiscs reconstitution the lipid composition can be manipulated using PC, PE or PS that are known to lack the activating effect and this should lead to the modification of the putative lipid density. I also think this section needs some improvement of the writing. The phrase "coupled binding of the lipid and cargo" is cryptic. What is the mechanism proposed? Please explain with more detail.

7. The human ATP13A2 has been shown to be N-glycosylated. Despite the fact that the exact sequence seems not conserved, there are no carbohydrates in CtPK9 that can explain the unexpected densities?

8. page 6. The authors identify a long continuous density in the E2Pinhb in the catalytic core of yPK9 interacting with several known functionally important P-ATPase motifs and assign it to the sequence of yPK9 prior to NTD. The authors speculate that this density corresponds to an auto-inhibitory interaction as it has been described in other P-ATPases. This interesting interpretation should be moved to Discussion since there are no yet experimental evidences of an autoinhibitory mechanism in P5B-ATPases. Alternatively the authors may want to consider the possibility of adding new experimental data i.e. showing that deletion of D74-G97 activates yPK9. Also the autoinhibition proposal shouldn't be restricted to yeast yPK9? There is no homologous segment in human PK9 except for a short N-terminal tail.

9. page7. The authors propose a reaction cycle for CtPK9 based in their suggestion that the N-terminus prevents the turnover and stabilize the enzyme in the E2Pinh. To substantiate their proposal the authors need to show that the CtYPK9 phosphoenzyme is mostly of the ADP-insensitive form. Although a mechanism like the one proposed is possible, it is not consistent with previous findings in the related ATP13A2 showing that during turn over the human ATP13A2 is mostly in the E1P form. Holemans T, Sørensen DM, van Veen S, Martin S, Hermans D, Kemmer GC, Van den Haute C, Baekelandt V, Günther Pomorski T, Agostinis P, Wuytack F, Palmgren M, Eggermont J, Vangheluwe P. A lipid switch unlocks Parkinson's disease-associated ATP13A2. Proc Natl Acad Sci U S A. 2015 Jul 21;112(29):9040-5. doi: 10.1073/pnas.1508220112. Epub 2015 Jul 1. PMID: 26134396; PMCID: PMC4517210. Are the authors are proposing a mechanism for the auto inhibited yeast PK9 different from that of human ATP13A2?

Minor points

1) Introduction section Paragraph "The yeast P5A-ATPase was recently determined to be a helix dislocase...." Only Ref. 6 is pertinent.

2) Please keep consistency in the nomenclature of P5A N terminal transmembrane helices as Ma and Mb. In the Introduction section are called MA and MB and in Results Ma and Mb.

Hugo Adamo, PhD.

Reviewer #2:

Remarks to the Author:

Gourdon and colleagues report the first cryoEM structure of Ypk9, a P5B-ATPase in yeast, in luminal-facing E2P, E2.Pi and autoinhibited states. The structures of Ypk9 provide novel insights, including its overall architecture, amphiphilic NTD helices which likely embedded in the membrane, and N-terminal auto-inhibition domain. They also provide polyamine-bound structures in the TM cleft in which conserved acidic and hydrophobic residues are located. Although E1 state structure and exit pathway of SPM are still elusive, present data will, no doubt, provide great contribution to the field. This reviewer hopes that the following comments improve the manuscript.

It is unclear whether E2.Pi(SPM) state is actually phosphorylated or not. Is it phosphorylated by carry-over ATP? Or no extra density observed at the phosphorylation site? Author wrote at Page4, the end of 1st para " stabilize an E2 state", which is confusing whether this structure represents SPM-bound E2.Pi state or E2 state.

It is unclear the reason why E2P(inhibit) is not capable for the polyamine binding. How much different the binding cavity structures between E2P(Inhibit), E2P* and E2.Pi(SPM or AIF/SPM) states? The architecture and dimension of SPM-binding pocket is not well presented. Although EM densities of SPM are shown, it is hardly recognized how SPM is accommodated in it, how it becomes available for SPM binding in E2.Pi state, what prevents SPM binding in E2P(inhibit) state. Followings are suggestions from this reviewer.

1. Membrane slice showing SPM-binding cavity with SPM density would be helpful to recognize the dimension of the binding site, and clarify whether the luminal gate is actually open or closed, instead of showing ribbon model alone as in Fig. 3C-E.
2. Although the rearrangement of TM helices is shown in cartoon models (Fig. 3B), it is difficult to recognize open/close of the luminal gate by showing only the side view. From the luminal view like Fig. S6 would help readers to recognize it. Also, it is pity that its atomic details are missing.
3. Distance of 1.3Å between SPM and Asp (Fig. 2EF) is too close, impossible for the non-covalent interaction. Authors should carefully re-modeling SPM binding mode.

Having essentially three different states with or without bound substrate in hand, these comparisons could explain an allosteric mechanism of how auto-inhibition occurred in the cytoplasmic domains prevents SPM binding at TM helices.

It is unclear whether phosphorylation (mimicked by BeF) or SPM-binding release the autoinhibition. In the model in Fig 4, autoinhibition region is removed at E2P* state. But on the other hands its EM density is, low contour level though, still visible in E2P* state in Fig. S9.

Considering about uphill transport of SPM, SPM occlusion seems required, as authors mentioned about

cargo occlusion of P1-P4 ATPases in E2-AIF (E2.Pi) state. But it seems not the case for P5B (and probably P5A as well) ATPase, as SPM seems accessible to the luminal solution in E2.Pi state. Please clarify whether SPM is occluded or not in E2.Pi state. Certainly, SPM can be seen from the luminal view, but it is unclear that the diameter of luminal gate opening is actually sufficient for the SPM dissociation (in this case SPM is bound, not occluded), or it is too narrow (in this case, SPM is occluded).

Maybe this is an open-ended question, if P5B-ATPase does not have a cargo occlusion state, how does this enzyme achieve up-hill transport? Alternatively, is it simply because the present structural analysis could not trap an occlusion structure?

Regarding autoinhibition domain, comparison between Drs2p and Ypk9 is missing. This reviewer will appreciate if authors mention the reason why E2P-like conformation is stabilized by autoinhibition? Is it because to increase a chance for SPM binding from the luminal side? Outward-facing conformation is easy to regulate inhibition/activation by SPM binding?

Minor comments

Main text should be carefully proofed. There are some errors including followings;

Page4 2nd para Missing “)” after Fig. S6)

Page 5 L2 Here in E2P and E2.Pi states, SPM is bound from the luminal solution to its binding site, and does not release to the cytosol. Therefore, “uptake” seems not suitable. Maybe “transport” or “SPM binding” are better explained.

P5 L6 “incapacity” seems not suitable. Because E2P(inhib) becomes E2P* in the presence of SPM, E2P(inhib) is capable for SPM binding, and induced conformational changes to become E2P* state.

P8 the last sentence before Summary

I could not catch the meaning of this sentence. Does it mean, for P1-3 type pathway, SPM need to release between TM1 and TM2, via TM4 unwinding, which is complicate and unlikely occurs?

Fig S8 Electron density map -> EM density map

Kazu Abe

Response to the comments obtained from the reviewers.

Remarks and questions from the reviewer are shown in black. Our responses are shown in green.

Revisions in the actual manuscript is highlighted in grey, of which some relate to *Nature Communications* format requests.

Reviewer #1 (Remarks to the Author):

What are the noteworthy results?

This paper describes four atomic resolution structures obtained by CryoEM of yPK9 from the yeast *C. thermophylus*, which is related to the human ATP13A2 protein associated with juvenile Parkinson's, and other neurological pathologies. P5B-ATPases have recently reported to transport spermidine and other polyamines from the lysosome to the cytosol. This transport specificity, unique among P-ATPases, correlates with structural features not conserved in other members of this superfamily.

- Will the work be of significance to the field and related fields? How does it compare to the established literature? If the work is not original, please provide relevant references.

The data presented in this manuscript is highly relevant since it is the first report of an atomic model structure of a P5B-ATPase. Based on the observations the authors propose several functional mechanisms that are worth to consider and will open the door to future experimentation.

- Does the work support the conclusions and claims, or is additional evidence needed?

The manuscript has structural information highly valuable in itself. The authors engage in several interpretations which need additional experimental evidence.

- Are there any flaws in the data analysis, interpretation and conclusions? - Do these prohibit publication or require revision?

I find no flaws in the data or interpretations although some of the data included show a poor fit to the EM map. I think the manuscript is a valuable contribution but may be improved by a Thoughtful revision.

- Is the methodology sound? Does the work meet the expected standards in your field?

The methodology is sound. One thing that surprises me is that according to the PDB validation a great number of residues (54% for the AF3/SPM structure) exhibit a poor fit to the EM map. This value is substantially lower for other CryoEM structures of P-ATPases that have been reported previously and somehow it contrasts with the reported 3.7 Å resolution.

- Is there enough detail provided in the methods for the work to be reproduced?

yes.

We thank the reviewer for the positive evaluation of our work – highly appreciated! As also outlined below, we have partly followed the advice to separate between results and discussion more clearly. Regarding the AIF3/SPM structure, we agree with the reviewer that this particular PDB left room for improvements. We have now replaced these data and the structure with freshly collected data and a new (almost identical) structure. The new PDB validation report is attached to this revision, suggesting a considerably better fit between the data and the model. All figures of the manuscript have been updated accordingly.

Major points.

1. Manuscript organization. The manuscript is organized in Abstract, Introduction, Results and Summary. As written the Results experimental observations are mixed with interesting but speculative interpretations without experimental support. I suggest the authors to make separate sections of Results with a detailed description of each new structure and a Discussion section containing the functional interpretations and possible implications of the new structural features.

Thank you for your suggestion to change the organization of the manuscript. We however prefer to keep Results and Discussion combined, which is rather common outline in Nature communications (e.g. <https://www.nature.com/articles/s41467-021-22290-1>, <https://www.nature.com/articles/s41467-021-21924-8>). However, we agree that speculations and interpretations need to be highlighted and separated from actual findings, and we have hence adapted the main text accordingly. Here are two examples “Based on these observations, we speculate that the affinity for SPM increases from the E2P^{inhib} to the E2P* and E2.P_i states.” and “We therefore hypothesize that the observed density relates to a lipid of the cytosolic membrane leaflet. The lipid may serve a regulatory function, linking the NTD and A-domain to TM1, 3 and 4.”

2. page 3 line 11 “In addition, an N-terminal domain (NTD) related to the one observed for the P5A-ATPases is detected”. Which is related NTD sequence of P5A and P5B? I don’t see there is much sequence conservation in this region.

We agree that there is little sequence conservation in the NTD region of P5A- and P5B-ATPases. However, the determined structure of the Ypk9 NTD harbours a seven-stranded β -barrel fold similar to that of Spf1 (e.g. PDB ID 6XMU), which leads us to conclude that the soluble parts of the NTD of P5A- and P5B-ATPases are similar. We therefore added the word “structurally” to the text: “In addition, an N-terminal domain (NTD) structurally related to the one observed for the P5A-ATPases is detected.” We also added an additional figure (Fig. S4), showing a structure-based alignment of the NTD of Ypk9 vs. Spf1.

3. page 3 line 18 “Consequently, the N-terminus prior to the NTD appears unique to the P5B-ATPases....” The authors define NTD as starting at CtPK9 residue S206 forward. What is the region prior to NTD the authors are referring to? ATP13A2 has only 29 residues. CtSpf1 has 18 residues. It seems to me that only the yeast yPK9 have a long segment previous to NTD.

In the E2P inhib structure, we modelled residues D74-G96 as a tail that interact tightly with the N- and P-domains (see Figs. 3G, S4 and S11). In addition, weak cryo-EM density is located near the membrane interface, which we expect will link between the modelled tail and S206 (see Fig. S10). These features together with the unmodelled residues in front of the tail constitute “the region prior to NTD”. The figure legends of Fig. S4 now reads: “The NTD domain harbors residues R204-G344 and is thus preceded by approximately 200 residues in Ypk9, of which D74-G96 are modelled.”

It is true that the sequence and even length of the N-termini, before residue S206 of Ypk9, is diverse (see Figs. S3, S4 and S11). However, the distance from the end of the tail to the first modelled residue of the NTD is 36 Å in the E2P inhib structure. Furthermore, the end of the autoinhibitory tail is surface exposed interacting with the P-domain only (not in-between domains), and hence likely not involved in the actual autoinhibition. If this stretch is not considered, the tail-to-NTD distance length is reduced to 30 Å and the size/length of the tail is 14 residues. The latter scenario would make a similar autoinhibitory mechanism in human ATP13A2 possible, but it is correct that autoinhibition using this mechanism cannot be omnipresent among P5B-ATPases. Accordingly, the main text has been revised: “However, the N-termini of P5B-ATPases are not conserved (Figs. S3 & S4), and such a mechanism would likely be limited to P5B-ATPases with an N-terminus of sufficient length, e.g. Ypk9.”

4. page 5 line 5 “This observation is congruent with polyamine binding incapacity for the E2P inhib state determined in the absence of SPM....” I believe this is an interesting interpretation that I would move to a Discussion section. There is no experimental data substantiating the lack of SPM binding to E2Pinhb.

Yes, we agree with the reviewer that there is no experimental data substantiating lack of SPM binding to E2Pinhb. As illustrated in figure 3C, we did not detect SPM density in the E2Pinhb state. The E2P* data set was generated in the presence of both SPM and BeF, and, if present at all, the observed SPM density was significantly weaker than for the E2.P_i states (Fig. 3D,E). We now added representations of the inner surface around the SPM binding pocket to Figs. S7 and S8. This suggests that the SPM binding pocket is not yet formed in the E2P inhib state. Nevertheless, we have revised the text, indicating that SPM binding cannot be excluded in the E2P inhib state: “The SPM binding pocket is not yet defined in the E2P^{inhib} conformation, where the luminal parts of TM1-2, TM3-4 and TM5-10 are further apart. Based on these observations, we speculate that the affinity for SPM increases from the E2P^{inhib} to the E2P* and E2.P_i states.” and also in the figure legend of Fig. S8: “While not supported by our data, SPM binding to the E2P states cannot be excluded.” See also the answer to one of the minor questions of reviewer 2.

5. page 5 .Second paragraph. The authors describe the structure and speculate about the possible function of NTD. It would help the reader if NTD would be better defined including a new figure showing exclusively the sequence and structural alignments of NTD of P5B and P5A (this will also help in Mayor point 2).

We thank the reviewer for this advice. We have included a new figure (Fig. S4) showing a structural alignment of Ypk9 vs. Spf1, and a sequence alignment of the NTD region. The region is also defined in the legend of Figure S4: “The NTD domain harbors residues R204-G344 and is thus preceded by approximately 200 residues in Ypk9, of which D74-G96 are modelled.”.

6. page5 .Third paragraph. The authors detect an unexpected density in E2Pi structure. Then the authors go on assuming that this density “relates to a lipid of the cytosolic membrane leaflet”. This section seems interesting but speculative in the absence of a better resolution. Experimental data indicating that the detected density is compatible with phosphatidic acid or PI(3,5)P2 needs to be provided. In addition to add support of the authors claim, during the nanodiscs reconstitution the lipid composition can be manipulated using PC, PE or PS that are known to lack the activating effect and this should lead to the modification of the putative lipid density. I also think this section needs some improvement of the writing. The phrase “coupled binding of the lipid and cargo” is cryptic. What is the mechanism proposed? Please explain with more detail.

We agree that the section is speculative concerning the origin of the detected feature. Our suggestion that the density belongs to a lipid of the cytosolic membrane leaflet is based on its position compared to the nanodisc and transmembrane domain, as apparent from the overview in Fig. 3F. In addition, the density is only present in the E2.P_i^{SPM} and E2.P_i^{AIF/SPM} data sets, indicative of a tighter interaction of the detected feature with the protein in this conformation. Because of the rearrangements of TM1,3,4 and NTD in the E2P → E2.P_i transition, the distance from the additional density to the NTD decreases. Interestingly, Holemans *et al.* (Proc Natl Acad Sci USA 2015;112, 9040–9045) revealed three regions of the NTD to be affected by PA and/or PI(3,5)P2. One of these three regions correspond to L248-P252 in Ypk9, including the conserved W250 in the membrane-dipping loop. Notably, L248-P252 represent the residues in the NTD that are closest to the detected additional density, and therefore we propose a possible connection of the additional density to lipid regulation.

For clarification, we have also attempted fitting of PC, PA and PI(3,5)P2 headgroups, respectively, into the additional density (Fig. S10B). However, due to the low resolution and uncertainty about the

nature of this density, we prefer not to model any ligand in the final structures. And indeed, it cannot be excluded that the feature belongs to for example detergent.

We have revised the text to: “Interestingly, two of these NTD regions are located in the membrane-dipping region of the Ypk9 NTD (L248-P252 and K257-K262), and the remaining portion is in close proximity to the observed density. We therefore hypothesize that the observed density relates to a lipid of the cytosolic membrane leaflet. The lipid may serve a regulatory function, linking the NTD and A-domain to TM1, 3 and 4. However, the presence of an unpredicted glycosylation site, or even a non-native detergent molecule cannot be excluded (alternative lipid effects via the highly electropositive region around the C-terminus are also possible, Fig. S9).”

7. The human ATP13A2 has been shown to be N-glycosylated. Despite the fact that the exact sequence seems not conserved, there are no carbohydrates in CtPK9 that can explain the unexpected densities?

We have assessed potential glycosylation sites of CtYpk9 using the GlycoPred and Prosite servers, and none of the suggested sites are close to the additional density adjacent to the NTD. In addition, the density was observed only in the E2.P_i states, which makes a glycosylation site less likely. However, it cannot be excluded that the observed density corresponds to a glycosylation site, and we therefore adapted the text as follows: “However, the presence of an unpredicted glycosylation site, or even a non-native detergent molecule cannot be excluded (alternative lipid effects via the highly electropositive region around the C-terminus are also possible, Fig. S9).”

8. page 6. The authors identify a long continuous density in the E2P_{inhib} in the catalytic core of yPK9 interacting with several known functionally important P-ATPase motifs and assign it to the sequence of yPK9 prior to NTD. The authors speculate that this density corresponds to an auto-inhibitory interaction as it has been described in other P-ATPases. This interesting interpretation should be moved to Discussion since there are no yet experimental evidences of an autoinhibitory mechanism in P5B-ATPases. Alternatively the authors may want to consider the possibility of adding new experimental data i.e. showing that deletion of D74-G97 activates yPK9. Also the autoinhibition proposal shouldn't be restricted to yeast yPK9? There is no homologous segment in human PK9 except for a short N-terminal tail.

We agree that the proposed mechanism would be substantiated by functional data. In this light, we have now generated a deletion Ypk9 form lacking the first 100 residues and compared the activity to the wild-type and a dead-mutant (the latter targeting the catalytical phosphorylation site, Asp680). Remarkably, we detected the anticipated stimulation of the activity, in agreement with the notion that the N-terminus indeed serves an inhibitory role. We have adapted the main text accordingly:” Indeed, a Ypk9 form lacking the first 100 residues, Ypk9 Δ_{N100} , also demonstrates higher activity in our in vitro assay (Fig. 2A). Thus, it appears as if the observed N-terminal feature indeed serves an auto-inhibitory role, perhaps exclusively stabilizing the E2P conformation, and that the P4- and P5B-ATPase regulatory mechanisms share common features .”. For clarification, we also added a structural comparison to ATP8A1 and Drs2p to Fig. S11, illustrating the structural overlap of the P4-ATPase autoinhibition domain with Ypk9 in the E2P^{inhib} state at the nucleotide binding site. The new functional data and this structural overlap lead us to conclude that the additional density in Ypk9 fulfils a similar function compared to the autoinhibition domain in P4-ATPases. In our opinion, the additional density, due to the tight interactions with all soluble domains, likely stabilizes the pump in its conformation.

As also outlined above, we agree that the N-terminus is not conserved among P5B-ATPases, and in some members, e.g. human ATP13A5, the N-terminus is too short to accommodate the density observed in the Ypk9 E2P^{inhib} data set. It may be that the autoinhibition mechanism is only present in certain P5B-ATPases such as Ypk9.

9. page7. The authors propose a reaction cycle for CtPK9 based in their suggestion that the N-terminus prevents the turnover and stabilize the enzyme in the E2P_{inh}. To substantiate their proposal the authors need to show that the CtYPK9 phosphoenzyme is mostly of the ADP-insensitive form.

Although a mechanism like the one proposed is possible, it is not consistent with previous findings in the related ATP13A2 showing that during turn over the human ATP13A2 is mostly in the E1P form. Holemans T, Sørensen DM, van Veen S, Martin S, Hermans D, Kemmer GC, Van den Haute C, Baekelandt V, Günther Pomorski T, Agostinis P, Wuytack F, Palmgren M, Eggermont J, Vangheluwe P. A lipid switch unlocks Parkinson's disease-associated ATP13A2. *Proc Natl Acad Sci U S A*. 2015 Jul 21;112(29):9040-5. doi: 10.1073/pnas.1508220112. Epub 2015 Jul 1. PMID: 26134396; PMCID: PMC4517210. Are the authors are proposing a mechanism for the auto inhibited yeast PK9 different from that of human ATP13A2?

Holemans *et al.* measured the effect of ADP or ATP on phospho-enzyme levels. In other P-type ATPases, the E1P → E2P transition is associated with major conformational changes in the soluble domains, and therefore only the E1P state is sensitive to ADP, as indicated by the reviewer. Holemans *et al.* observed a rapid decrease in phospho-enzyme levels with ADP (but not ATP), and therefore concluded that ATP13A2 accumulates in E1P. However, these data do not exclude that ATP13A2 accumulates in E2P instead of E1P. We note that several of the conformations resolved for Spfl align poorly with the equivalent conformations of SERCA, which may hint at considerable conformational differences between P5 and more classical P-type ATPase, with similar E1P and E2P states. If we instead assume the overall cycle is maintained among P-type ATPases, then binding of the inhibitory loop in an E1P conformation is unlikely, as the N-domain then would be closer to the phosphorylation motif of the P-domain, precluding the currently identified structural arrangement. As also mentioned in major points 2 and 8, it is likely that the proposed autoinhibition mechanism is not omnipresent among P5B-ATPases, due to the substantially different lengths of the N-termini. Therefore, ATP13A2 may accumulate in E1P (as suggested by Holemans *et al.*), and Ypk9 in E2P as also indicated by the reviewer. Nevertheless, we have expanded the text with: “The aforementioned notion is not in disagreement with the observed “resting“ autophosphorylated configuration detected in ATP13A2¹⁴”.

Minor points

1) Introduction section Paragraph “The yeast P5A-ATPase was recently determined to be a helix dislocase....” Only Ref. 6 is pertinent.

References 4 and 5 have been removed from the manuscript.

2) Please keep consistency in the nomenclature of P5A N terminal transmembrane helices as Ma and Mb. In the Introduction section are called MA and MB and in Results Ma and Mb.

We changed the nomenclature in the introduction to Ma and Mb.

Reviewer #2 (Remarks to the Author):

Gourdon and colleagues report the first cryoEM structure of Ypk9, a P5B-ATPase in yeast, in luminal-facing E2P, E2.Pi and autoinhibited states. The structures of Ypk9 provide novel insights, including its overall architecture, amphiphilic NTD helices which likely embedded in the membrane, and N-terminal auto-inhibition domain. They also provide polyamine-bound structures in the TM cleft in which conserved acidic and hydrophobic residues are located. Although E1 state structure and exit pathway of SPM are still elusive, present data will, no doubt, provide great contribution to the field. This reviewer hopes that the following comments improve the manuscript.

We thank the reviewer for this very positive evaluation of our work and for the suggestions for how to improve the manuscript.

It is unclear whether E2.Pi(SPM) state is actually phosphorylated or not. Is it phosphorylated by carry-over ATP? Or no extra density observed at the phosphorylation site? Author wrote at Page4, the end of 1st para “ stabilize an E2 state”, which is confusing whether this structure represents SPM-bound E2.Pi state or E2 state.

The E2.P_i^{SPM} structure was generated in the absence of phosphate mimics, such as AlF or BeF, but in presence of SPM. The resulting conformation is E2.P_i, which we conclude based on the similarity to the E2.P_i^{AlF/SPM}, and difference to the E2P and E2P* data. We changed the text from “E2” to “E2.P_i” on page 4, end of 1st paragraph. We agree that this conformation should be phosphorylated (in the process if being dephosphorylated). However, our data at the phosphorylation site are unfortunately inconclusive, and it is therefore difficult to make any firm statement as to whether Ypk9 is phosphorylated by carry-over ATP or not. It cannot be excluded that the SPM-bound E2.P_i state has been obtained through reverse turn-over, stimulated by SPM (but hence not phosphorylated).

It is unclear the reason why E2P(inhibit) is not capable for the polyamine binding. How much different the binding cavity structures between E2P(Inhibit), E2P* and E2.Pi(SPM or AlF/SPM) states? The architecture and dimension of SPM-binding pocket is not well presented. Although EM densities of SPM are shown, it is hardly recognized how SPM is accommodated in it, how it becomes available for SPM binding in E2.Pi state, what prevents SPM binding in E2P(inhibit) state. Followings are suggestions from this reviewer.

1. Membrane slice showing SPM-binding cavity with SPM density would be helpful to recognize the dimension of the binding site, and clarify whether the luminal gate is actually open or closed, instead of showing ribbon model alone as in Fig. 3C-E.

We thank the reviewer for these constructive suggestions that indeed improve the manuscript. Accordingly, we adapted Fig. 3C-E, Fig. S7 and added Fig. S8. As apparent from the side-views in Fig. S7, the luminal opening is actually wider in the E2P and E2P* states. But in the E2P state, the inner surface is not continuous, so the cavity appears closed as seen from the luminal side. E2P* represents a transition state, but the actual SPM pocket forms only in the E2.P_i conformations. In Fig. 3C-E we now present slices through the surface together with SPM density and a model in cartoon representation. Atomic details are illustrated in Fig. S8. We also adapted the text on page 5, paragraph 1: “However, while we do not detect polyamine indications in the E2P^{inhib} structure, poor cryo-EM density for SPM, somewhat shifted towards the luminal side compared to the E2.P_i structures, is available for E2P* (Fig. 3C-E). The SPM binding pocket is not yet defined in the E2P^{inhib} conformation, where the luminal parts of TM1-2, TM3-4 and TM5-10 are further apart. Based on these observations, we speculate that the affinity for SPM increases from the E2P^{inhib} to the E2P* and E2.P_i states. The structural changes that orchestrate the SPM pocket are coupled to a rotation of the A-domain, leading to a 6 Å movement of the TGES dephosphorylation loop as observed in other P-type ATPases, and a similar movement of the adjacent NTD (Fig. 3A).”.

2. Although the rearrangement of TM helices is shown in cartoon models (Fig. 3B), it is difficult to recognize open/close of the luminal gate by showing only the side view. From the luminal view like Fig. S6 would help readers to recognize it. Also, it is pity that its atomic details are missing.

This question has been answered in the previous response. We have added atomic details to Fig. S8.

3. Distance of 1.3Å between SPM and Asp (Fig. 2EF) is too close, impossible for the non-covalent interaction. Authors should carefully re-modelling SPM binding mode.

We thank the reviewer for this request. We re-modelled SPM, and the distance to D1184 is now 2.2 Å.

Having essentially three different states with or without bound substrate in hand, these comparisons could explain an allosteric mechanism of how auto-inhibition occurred in the cytoplasmic domains prevents SPM binding at TM helices.

It is unclear whether phosphorylation (mimicked by BeF) or SPM-binding release the autoinhibition. In the model in Fig 4, autoinhibition region is removed at E2P* state. But on the other hands its EM density is, low contour level though, still visible in E2P* state in Fig. S9.

We thank the reviewer for raising this matter - we agree that this is an enigmatic topic. We note it is possible that we observe a mixture of two different populations in the E2P* data set, i.e. the E2P state caused by BeF, and the E2.Pi state triggered by SPM, as we observe weak density for both SPM and the autoinhibition loop in the E2P* state, in agreement with the statement of the reviewer. However, because the E2P* and E2P^{inhib} states are highly homologous this is unlikely. Our view is that it is unlikely that phosphorylation triggers release of the autoinhibitory loop. We base this 1) on the fact that the E2P^{inhib} state was generated in the presence of the phosphate mimic BeF, and 2) that the E2P^{inhib} and E2P* conformations are similar to the fully phosphorylated E2P conformation of SERCA, suggesting that phosphorylation has been completed in the observed E2P^{inhib} structure. Instead, we favour the interpretation that it is SPM binding and/or dephosphorylation that trigger release of auto-inhibition. The figure legend of Fig. 4 has been changed accordingly: “Polyamine binding and the coupled dephosphorylation lead to release of the auto-inhibitory loop, and the formation of a binding pocket between TM 2, 4 and 6 (E2P*).” We also note that for the flippase ATP8A2, it is the phosphorylation of the C-terminal autoinhibitory domain by Cam-kinase that releases the autoinhibitory loop. A similar release mechanism cannot be excluded here, but we do not have any data pointing in this direction.

Considering about uphill transport of SPM, SPM occlusion seems required, as authors mentioned about cargo occlusion of P1-P4 ATPases in E2-AIF (E2.Pi) state. But it seems not the case for P5B (and probably P5A as well) ATPase, as SPM seems accessible to the luminal solution in E2.Pi state. Please clarify whether SPM is occluded or not in E2.Pi state. Certainly, SPM can be seen from the luminal view, but it is unclear that the diameter of luminal gate opening is actually sufficient for the SPM dissociation (in this case SPM is bound, not occluded), or it is too narrow (in this case, SPM is occluded).

SPM is surface-accessible and hence not occluded in the E2.Pi state, but highly stabilized in its position due to a tight interaction-network with conserved electronegative residues in the binding pocket. Due to the absence of structural information on E1 states, we cannot draw any conclusions about the SPM release pathway. We believe a SPM transport mechanism without an occluded state indeed is possible, as the reviewer suggests. The text has been revised accordingly: “Moreover, we note that the TM2, 4 and 6 cargo-binding area overlaps well with the one observed in both P5A-ATPases, Spfl (PDB-ID 6XMU), and P4-ATPases, ATP8A1-CDC50a (6K7M) and ATP11C-CDC50a (6LKN & 7BSU) (Fig. 2D)^{4,18,20,21}, and that these separate cargoes are less occluded as compared to P2- and P3-ATPases.” and “However, while we do not detect polyamine indications in the E2P^{inhib} structure, poor cryo-EM density for SPM, somewhat shifted towards the luminal side compared to the E2.Pi structures, is available for E2P* (Fig. 3C-E). The SPM binding pocket is not yet defined in the E2P^{inhib} conformation, where the luminal parts of TM1-2, TM3-4 and TM5-10 are further apart. Based on these observations, we speculate that the affinity for SPM increases from the E2P^{inhib} to the E2P* and E2.Pi states.”. For details regarding SPM binding, please see our answers to question 1.

Maybe this is an open-ended question, if P5B-ATPase does not have a cargo occlusion state, how does this enzyme achieve up-hill transport? Alternatively, is it simply because the present structural analysis could not trap an occlusion structure?

Due to the size of SPM, it is possible that no occluded state is present. As mentioned above, P5A-ATPases do not have an occluded state, and only the lipid headgroup is membrane-embedded in P4-ATPases. We believe that SPM is sufficiently stabilized in the E2.P_i state so that back-release to the lumen is prevented. The text has been revised accordingly: “Moreover, we note that the TM2, 4 and 6 cargo-binding area overlaps well with the one observed in both P5A-ATPases, Spf1 (PDB-ID 6XMU), and P4-ATPases, ATP8A1-CDC50a (6K7M) and ATP11C-CDC50a (6LKN & 7BSU) (Fig. 2D)^{4,18,20,21}, and that these separate cargoes are less occluded as compared to P2- and P3-ATPases.”.

Regarding autoinhibition domain, comparison between Drs2p and Ypk9 is missing. This reviewer will appreciate if authors mention the reason why E2P-like conformation is stabilized by autoinhibition? Is it because to increase a chance for SPM binding from the luminal side? Outward-facing conformation is easy to regulate inhibition/activation by SPM binding?

We agree with the reviewer that it is reasonable to compare with Drs2p. This (and a comparison to ATP8A1) has now been included in Fig. S11, showing partially overlapping inhibitory regions between Ypk9 and Drs2p. The reviewer is likely correct that autoinhibition provides additional SPM sensitivity, and an increased chance for SPM binding from the luminal side, and in this way, SPM may also serve as a molecular switch. SPM can likely not serve this role if an E1 state was to be inhibited.

Minor comments

Main text should be carefully proofed. There are some errors including followings;

Page4 2nd para Missing “)” after Fig. S6)

Thank you very much for pointing this out. We have now carefully proofread the main text.

Page 5 L2 Here in E2P and E2.P_i states, SPM is bound from the luminal solution to its binding site, and does not release to the cytosol. Therefore, “uptake” seems not suitable. Maybe “transport” or “SPM binding” are better explained.

We have now revised the manuscript to: “Comparison of the structures reveal major rearrangements of the TM helices linked to SPM binding (Fig. 3A,B).”

P5 L6 “incapacity” seems not suitable. Because E2P(inhib) becomes E2P* in the presence of SPM, E2P(inhib) is capable for SPM binding, and induced conformational changes to become E2P* state.

We thank the reviewer for this insightful comment. We have changed the text to: “This observation is congruent with increasing affinity for SPM from the E2P^{inhib} (determined in the absence of SPM) to the E2P* (determined in the presence of SPM) and E2.P_i states (with bound SPM).”

P8 the last sentence before Summary

I could not catch the meaning of this sentence. Does it mean, for P1-3 type pathway, SPM need to release between TM1 and TM2, via TM4 unwinding, which is complicate and unlikely occurs?

The intention with the sentence present in the first version of the manuscript was that in P1-3-ATPases, ion uptake is accomplished between TM 1,3 and 4. If then Ypk9 used a similar pathway to P1-3-ATPases, it would release SPM in a similar manner, i.e. between TM 1,3 and 4. With the available data this appears unlikely, as TM4 blocks this putative SPM release pathway, and larger conformational changes that expected would be required, which we now have highlighted in the main

text: “This would however require a rather complex release mechanism, including re-arrangement of TM4 relative to TM1 to permit the polyamine to reach TM1, and we therefore believe that TM1 fulfils a role in lipid regulation rather than in cargo release.”

Fig S8 Electron density map -> EM density map

We thank the reviewer for pinpointing this error. We have changed the figure legend accordingly.

Reviewers' Comments:

Reviewer #1:

Remarks to the Author:

In the new version of this manuscript, the authors have appropriately adapted the text, made corrections, and incorporated new data. The hypotheses are now better distinguished from the results. The new supplemental Figure 4 allows a better understanding of the author's argument about the NTD region in P5B and P5A ATPases. The possible divergence between the proposed autoinhibitory segments in the functional consequences of different P5B ATPases (yeast and human) is now explicitly mentioned. Although I find it not very likely, the author's suggestion about differences in the reaction cycle of yPK9 and ATP13A2 (E2P vs E1P) is okay. This version of the manuscript incorporates new experimental data showing that the ATPase activity of mutant Ypk9 Δ N100 is somewhat higher (about 1.6 x?) than the WT, a fact that agrees with the author's proposed mechanism of autoinhibition of the yPK9. Just for curiosity, I would be eager to know the activity of this mutant in the absence of SPM.

I have no further criticisms and I find the manuscript suitable for publication.

Hugo Adamo

Reviewer #2:

Remarks to the Author:

The manuscript is significantly improved. Now polyamine binding site structure is well presented, and answers from authors are convincing. I have no further comments on this manuscript, and thus recommend this manuscript for the publication in Nature Communications.

Kazuhiro Abe

Regarding information requested “out of curiosity” from reviewer 1.

We have now measured the Ypk9- Δ 100 deletion activity in the absence of the SPM and the results suggest that the form presents relative high activity without the inhibition loop even without supplemented SPM (see panel A below). The activity of Ypk9- Δ 100 can also be inhibited by the classical inhibitors BeF3 and NaVO4 both in the absence/presence of supplemented SPM (see indicative data in panel B, based on a single measurement only).